# Circulating fatty acids and risk of hepatocellular carcinoma and chronic liver disease mortality in the UK Biobank

Zhening Liu[1,4], Hangkai Huang[1,4], Jiarong Xie[1,2,4], Yingying Xu[3] & Chengfu Xu [1] ✉

Nuclear magnetic resonance (NMR)-based plasma fatty acids are objective biomarkers of many diseases. Herein, we aim to explore the associations of NMR-based plasma fatty acids with the risk of hepatocellular carcinoma (HCC) and chronic liver disease (CLD) mortality in 252,398 UK Biobank participants. Here we show plasma levels of n-3 poly-unsaturated fatty acids (PUFA) and n-6 PUFA are negatively associated with the risk of incident HCC [$HR_{Q4vsQ1}$: 0.48 (95% CI: 0.33–0.69) and 0.48 (95% CI: 0.28–0.81), respectively] and CLD mortality [$HR_{Q4vsQ1}$: 0.21 (95% CI: 0.13–0.33) and 0.15 (95% CI: 0.08–0.30), respectively], whereas plasma levels of saturated fatty acids are positively associated with these outcomes [$HR_{Q4vsQ1}$: 3.55 (95% CI: 2.25–5.61) for HCC and 6.34 (95% CI: 3.68–10.92) for CLD mortality]. Furthermore, fibrosis stage significantly modifies the associations between PUFA and CLD mortality. This study contributes to the limited prospective evidence on the associations between plasma-specific fatty acids and end-stage liver outcomes.

Primary liver cancer ranks as the seventh most commonly occurring cancer in the world and the second most common cause of cancer mortality[1]. Hepatocellular carcinoma (HCC) is the dominant type of primary liver cancer and has become an important cause of cancer-related deaths in the UK[2]. Most HCCs occur in the context of chronic liver disease (CLD), such as viral hepatitis, metabolic dysfunction-associated steatotic liver disease (MASLD), and alcoholic fatty liver disease[3]. In addition, CLD leads to 2 million deaths worldwide each year and is therefore a notable public health concern[4]. Both HCC and CLD deaths represent advanced liver disease. The growing burden related to HCC and CLD emphasizes the importance of identifying people at high risk and taking measures as soon as possible[5,6].

Fatty acids are the main components of circulating lipids, which can be present in a free state (also known as non-esterified fatty acids) or combine with glycerol or cholesterol to form esters. According to the number and position of carbon–carbon double bonds, fatty acids can be broadly categorized into saturated fatty acids (SFA), mono-unsaturated fatty acids (MUFA), and poly-

unsaturated fatty acids (PUFA, which can be further categorized into n-3 and n-6 by the position of the first double bond in relation to the omega carbon)[7].

Overall, evidence regarding specific dietary fatty acids in relation to incident HCC remains inconsistent. It has been reported that n-6 PUFA and SFA intake displayed a significantly positive association with HCC risk[8,9], whereas no risk associations of HCC were detected for PUFA or SFA intake in a European cohort[10]. Furthermore, in an analysis of data from 2 large US cohort studies, increasing consumption of MUFA and PUFA, including both n-3 and n-6 PUFA, was associated with a lower risk of developing HCC[11]. Correspondingly, in a hospital-based case–control study, inverse associations of MUFA and long-chain n-3 PUFA intake with HCC were found[12]. Nevertheless, information based on dietary recalls inevitably introduces bias, while fatty acids in circulation are an attractive source of biomarkers[13]. In addition, the plasma lipidome closely parallels liver lipidome changes[14], and circulating fatty acids have also been found to reflect the composition of liver triglycerides,

[1]Department of Gastroenterology, Zhejiang Provincial Clinical Research Center for Digestive Diseases, the First Affiliated Hospital, Zhejiang University School of Medicine, Hangzhou 310003, China. [2]Department of Gastroenterology, the First Affiliated Hospital of Ningbo University, Ningbo 315010, China. [3]Department of Geriatrics, the Third People's Hospital of Yuyao, Yuyao 311101, China. [4]These authors contributed equally: Zhening Liu, Hangkai Huang, Jiarong Xie. ✉e-mail: xiaofu@zju.edu.cn

which makes it easier to explore the role of hepatic fatty acids compared with collecting human tissues[15]. Zhou et al. analyzed serum metabolomics and revealed the deregulation of FA metabolism in HCC and CLD[16]. A blood lipidomic study showed the importance of an increased ratio of long-chain n-6 PUFA over n-3 PUFA for HCC risk in both human samples and a mouse tumor model[17]. Besides, in 116 subjects of Hispanic descent, plasma concentrations of very long chain SFA and very long chain n-3 PUFA were significantly reduced in patients with HCC[18]. However, these studies have relatively small sample sizes (~100), thus it is necessary to examine the association of HCC with subtypes of circulating fatty acids in large cohorts. Furthermore, although evidence is accumulating on fatty acids and MASLD progression[19], the specific subtype of fatty acids that influences mortality from CLD remains poorly understood.

In addition to dietary consumption, the levels of circulating fatty acids also depend on endogenous synthesis and metabolism[20,21]. In individuals of European descent, the polymorphism of fatty acid desaturases (*FADS1/2* and *SCD*) and fatty acid elongases (*ELOVL2*) have been found to influence the levels of MUFA, SFA[20], as well as PUFA[21,22]. However, it is unknown whether these variants could modify the effects of specific fatty acids on the liver related outcomes.

Liver fibrosis is a common pathological feature of CLD, characterized by the formation of a fibrous scar[23]. Fibrosis stage is the most important in terms of determining disease progression to liver-associated complications and mortality[24]. Early stage of fibrosis is usually asymptomatic, nevertheless, once it progresses to the advanced stage or cirrhosis, the prognosis is always poor[25]. Thus, it is necessary to examine whether the associations between fatty acids and liver related outcomes vary by fibrosis stages.

Here, we show that plasma levels of n-3 PUFA and n-6 PUFA are negatively associated with the risk of incident HCC and CLD mortality, whereas plasma levels of SFA are positively associated with these outcomes. The plasma level of MUFA is only associated with a higher risk of CLD mortality. The present findings suggest that plasma fatty acids levels have important predictive value for severe liver outcomes.

## Results
### Population characteristics
The characteristics of all participants by quartiles of fatty acids are summarized in Supplementary Table 5. At baseline, participants with higher n-3 PUFA levels tended to be older, female, non-White, less deprived, highly educated, have more family income, never smoked, drink more frequently, and have a healthier diet. In addition, they had a lower waist circumference, body mass index (BMI), blood platelet count, alanine aminotransferase (ALT) level, and aspartate aminotransferase (AST) level, but a greater level of plasma triglycerides and total fatty acids. Participants with higher SFA levels were more likely to be older, White, less educated, current smokers, alcohol drinkers, and those with poor diet quality, but had less family income and less activity. Additionally, they had a higher waist circumference, BMI, serum ALT, AST, plasma triglycerides, total cholesterol, and total fatty acids levels. As for detailed dietary factors (Supplementary Fig. 3), individuals with higher plasma n-3 PUFA levels consumed more fruits, vegetables, whole grains, and fish but less refined grains and processed meats. On the contrary, those with higher plasma SFA levels consumed fewer fruits, vegetables, whole grains, and vegetable oils but more refined grains, processed meats, and unprocessed red meats.

The Spearman correlations between specific plasma fatty acids are summarized in Supplementary Table 6. Plasma levels of n-3 PUFA had weak negative correlations with n-6 PUFA ($r_s = -0.137$), MUFA ($r_s = -0.278$) and SFA ($r_s = -0.137$). Plasma levels of n-6 PUFA had strong negative correlations with MUFA ($r_s = -0.750$) and SFA ($r_s = -0.646$). There was a weak positive correlation between plasma SFA and MUFA ($r_s = 0.265$).

### Crude morbidity of HCC and mortality of CLD
During a median follow-up of 13.8 years (~3,400,000 person-years), 273 cases of HCC and 244 cases of fatal CLD were identified. The crude morbidity of HCC and mortality of CLD across specific fatty acids levels as well as the Kaplan–Meier curves are shown in Supplementary Figs. 4–5. For HCC, the highest crude incidence rate was found in the lowest quartile of plasma n-3 PUFA and n-6 PUFA levels but the highest quartile of plasma MUFA and PUFA levels. For CLD mortality, similar associations were also found (all with $P < 0.001$).

### Associations between plasma fatty acids and incident HCC risk
The heatmaps (Fig. 1) present the HRs of participants in quartiles 2–4 compared with the lowest quartile, with actual metrics included in Supplementary Table 7. In model 1, higher plasma proportions of n-3 and n-6 PUFA were associated with lower HCC risk, whereas SFA and MUFA were associated with higher risk. These relationships (except MUFA) remained strong in the fully adjusted model. For n-3 PUFA, compared with the lowest quartile, the multivariable HRs (95% CI) of incident HCC in quartiles 2–4 were 0.67 (0.49–0.92), 0.60 (0.43–0.83), and 0.48 (0.33–0.69), $P_{trend} < 0.001$. For n-6 PUFA, the multivariable HRs (95% CI) in quartiles 2–4 were 0.63 (0.43–0.92), 0.56 (0.35–0.89), and 0.48 (0.28–0.81), $P_{trend} = 0.018$. For SFA, the multivariable HRs (95% CI) in quartiles 2–4 were 1.81 (1.12–2.93), 3.26 (2.08–5.10), and 3.55 (2.25–5.61), $P_{trend} < 0.001$. Besides, for MUFA, only a positive tendency remained ($P_{trend} = 0.037$). Additionally, linoleic acid (LA) did not significant associate with the risk of HCC ($P_{trend} = 0.617$), and docosahexaenoic acid (DHA) only showed a negative tendency ($P_{trend} = 0.042$) in the fully adjusted model. n-6/n-3 showed a positive association with HCC [HR$_{Q4vsQ1}$: 1.95 (1.35–2.82), $P_{trend} < 0.001$], but MUFA/SFA showed a negative association [HR$_{Q4vsQ1}$: 0.48 (0.31–0.76), $P_{trend} = 0.008$, Supplementary Table 8].

### Associations between plasma fatty acids and risk of CLD mortality
For CLD mortality, the HRs and 95% CIs of participants in quartiles 2–4 compared with the lowest quartile are reported in Fig. 2 and Supplementary Table 9. Plasma n-3 PUFA and n-6 PUFA were negatively associated with CLD mortality; in contrast, plasma MUFA and SFA were related to higher CLD mortality in the minimally adjusted model. Similar results were still found after further adjustment for other demographic characteristics, lifestyle factors, and other plasma lipid parameters. Compared with the lowest quartile, the multivariable HRs in the highest categories were 0.21 (95% CI: 0.13–0.33) for n-3 PUFA ($P_{trend} < 0.001$), 0.15 (95% CI: 0.08–0.30) for n-6 PUFA ($P_{trend} < 0.001$), 3.81 (95% CI: 2.03–7.16) for MUFA ($P_{trend} < 0.001$), and 6.34 (95% CI: 3.68–10.92) for SFA ($P_{trend} < 0.001$). Correspondingly, LA and DHA were also significantly associated with a lower risk of CLD mortality [HR$_{Q4vsQ1}$: 0.35 (0.20–0.63) and 0.24 (0.14–0.39)]. n-6/n-3 had a positive association [HR$_{Q4vsQ1}$: 3.61 (2.39–5.45), $P_{trend} < 0.001$], but MUFA/SFA had a negative association [HR$_{Q4vsQ1}$: 0.49 (0.30–0.81), $P_{trend} = 0.005$, Supplementary Table 10]. When we stratified the causes of CLD by alcoholic liver diseases, liver fibrosis or cirrhosis, and MASLD, similar associations of specific fatty acids with alcoholic liver diseases and liver fibrosis or cirrhosis mortality were observed. However, only the positive association of SFA with MASLD morality was found (Supplementary Table 11).

### Modification effect of fibrosis stage
We next investigated a possible interaction between fibrosis stage and specific fatty acids in terms of the outcomes. As expected, a higher fibrosis risk was associated with a higher risk of HCC or CLD mortality ($P_{trend} < 0.001$, Supplementary Fig. 6). After stratifying associations of specific fatty acids with incident HCC by fibrosis stage (by FIB-4), no interactions were detected between specific fatty acids and fibrosis stage (Fig. 3a). As for CLD mortality,

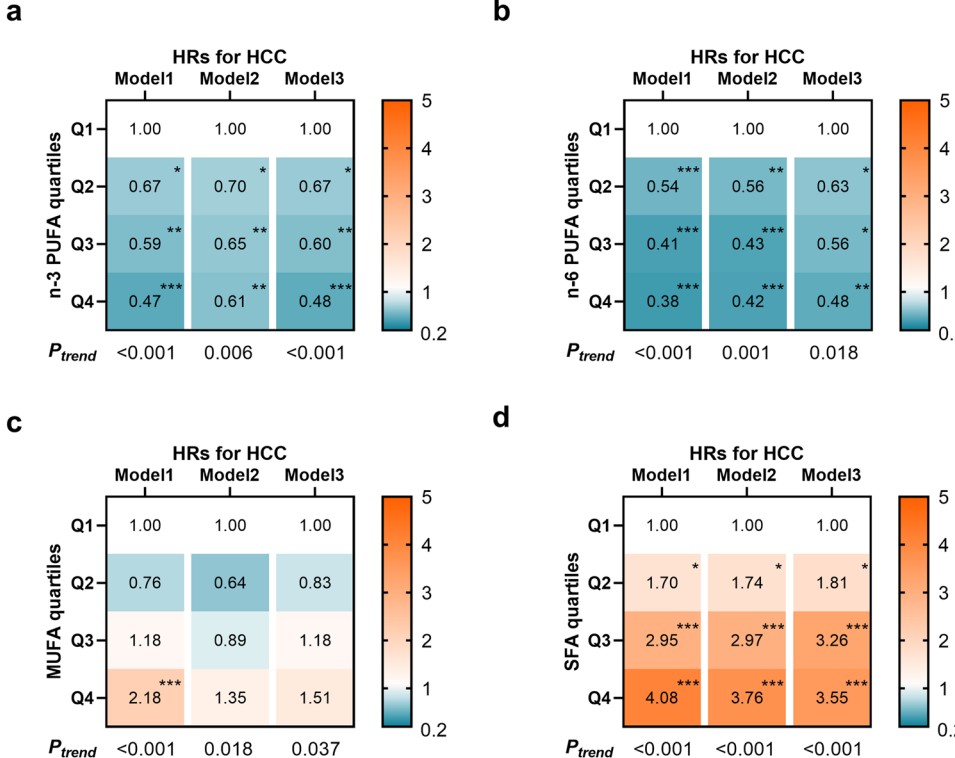

**Fig. 1 | Associations between plasma n-3 PUFA, n-6 PUFA, MUFA, SFA levels and incident HCC risk (*n* = 252,398 participants).** The heatmaps shown the HRs of **a** n-3 PUFA, **b** n-6 PUFA, **c** MUFA, and **d** SFA with incident HCC. Multivariable Cox proportional hazard models were used: Model 1 was adjusted for age, sex, and ethnicity. Model 2 was adjusted for model 1 plus BMI, waist circumference, Townsend deprivation index, education level, household income, self-reported smoking status, self-reported frequency of alcohol intake, physical activity, diet quality score, baseline hypertension, baseline diabetes, and baseline dyslipidemia.

Model 3 was adjusted for model 2 plus total cholesterol level, triglycerides level, total fatty acids level, serum ALT level, serum AST level, and blood platelet count. Quartiles of n-3 PUFA (% TFA): Q1, <3.3; Q2, 3.3–4.2; Q3, 4.2–5.2; Q4, >5.2. Quartiles of n-6 PUFA (% TFA): Q1, <35.7; Q2, 35.7–38.4; Q3, 38.4–40.4; Q4, >40.4. Quartiles of MUFA (% TFA): Q1, <21.8; Q2, 21.8–23.5; Q3, 23.5–25.4; Q4, >25.4. Quartiles of SFA (% TFA): Q1, <32.7; Q2, 32.7–33.9; Q3, 33.9–35.2; Q4, >35.2. Source data are provided as a Source Data file.

significant interactions were found between n-3 PUFA, n-6 PUFA and fibrosis stage ($P_{interaction}$ = 0.007 and 0.029, respectively). Among participants at high fibrosis risk (FIB-4 > 2.67), pronounced associations were found for n-3 PUFA [HR$_{1-SD}$ (95% CI): 0.41 (0.31–0.55)] and n-6 PUFA [HR$_{1-SD}$ (95% CI): 0.34 (0.23–0.51), Fig. 3b] compared to those at low fibrosis risk (FIB-4 < 1.30).

**Modification effect of polygenic risk scores (PRS) or other specific SNPs**

For PRS, higher genetic susceptibility was associated with a higher risk of HCC and CLD mortality ($P_{trend}$ < 0.001 and $P_{trend}$ = 0.023, respectively. Supplementary Fig. 7a). Significant interaction of plasma n-6 PUFA with PRS on HCC risk ($P_{interaction}$ = 0.019) were detected. In category analysis, the inverse associations of plasma n-6 PUFA with HCC were stronger among participants with higher PRS [HR$_{1-SD}$ (95% CI): 0.58 (0.39–0.84), Fig. 4a]. However, no significant interactions were observed between the other plasma fatty acids and PRS for HCC and CLD mortality (Fig. 4a, b). In addition, neither the main effects of rs2236212, rs3734398, rs174547, rs603424, and rs102275 genotypes with the risk of HCC and CLD mortality (Supplementary Fig. 7b-7f) nor the interaction effects between specific plasma fatty acids and the genotypes of above SNPs were observed (Figs. 4c–i).

**Sensitivity analyses and subgroup analyses**

We conducted several sensitivity analyses to examine the robustness of the findings. When we excluded the first 2 years of follow-up, and further adjusted for lipid-lowering medication, PRS, *FADS1/2* genotype,

or the remaining plasma fatty acids, the observed associations of specific fatty acids with incident HCC and CLD mortality remained unchanged (Supplementary Tables 12, 13).

In subgroup analyses, the negative associations of plasma n-3 PUFA with HCC risk (Supplementary Table 14) were more pronounced among those less than 60 years ($P_{interaction}$ = 0.034), and the association of plasma n-3 PUFA with CLD mortality were more pronounced among those over 60 years ($P_{interaction}$ = 0.041) and less social deprived ($P_{interaction}$ = 0.010). For n-6 PUFA (Supplementary Table 15), the negative associations with HCC risk were more pronounced among those who were older ($P_{interaction}$ = 0.003), and the negative associations with CLD mortality were more pronounced among men ($P_{interaction}$ = 0.001), those with BMI less than 25 kg/m² ($P_{interaction}$ < 0.001), and those drank more frequently ($P_{interaction}$ < 0.001). In addition, we observed stronger positive associations of plasma MUFA with HCC risk among older people ($P_{interaction}$ = 0.003), and stronger positive associations of MUFA with CLD mortality among men ($P_{interaction}$ = 0.006), those with BMI less than 25 kg/m² ($P_{interaction}$ = 0.020), those drank more frequently ($P_{interaction}$ = 0.002), and those without a history of diabetes ($P_{interaction}$ = 0.044, Supplementary Table 16). Additionally, the positive associations of plasma SFA and CLD mortality were more evident among men ($P_{interaction}$ = 0.002, Supplementary Table 17).

**Discussion**

In this large community-based cohort study, plasma levels of n-3 PUFA and n-6 PUFA were negatively associated with the risk of incident HCC and CLD mortality, whereas plasma levels of SFA were positively associated with these outcomes. The plasma level of

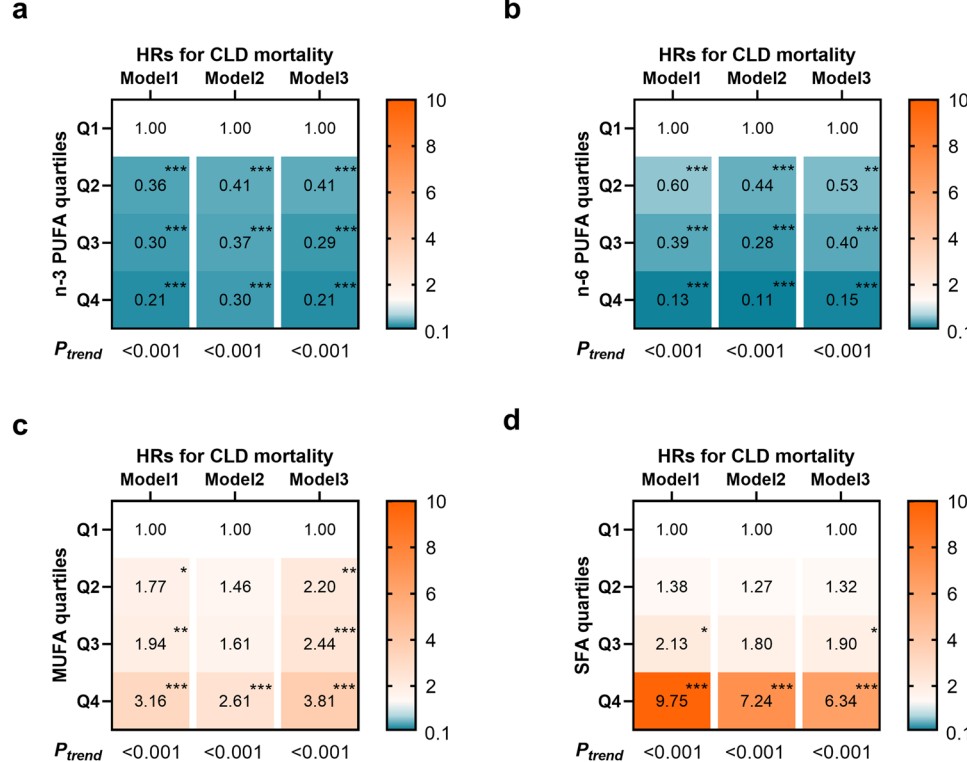

Fig. 2 | **Associations between plasma n-3 PUFA, n-6 PUFA, MUFA, SFA levels and risk of CLD mortality (*n* = 252,398 participants).** The heatmaps shown the HRs of **a** n-3 PUFA, **b** n-6 PUFA, **c** MUFA, and **d** SFA with CLD mortality. Multivariable Cox proportional hazard models were used: Model 1 was adjusted for age, sex, and ethnicity. Model 2 was adjusted for model 1 plus BMI, waist circumference, Townsend deprivation index, education level, household income, self-reported smoking status, self-reported frequency of alcohol intake, physical activity, diet quality score, baseline hypertension, baseline diabetes, and baseline dyslipidemia.

Model 3 was adjusted for model 2 plus total cholesterol level, triglycerides level, total fatty acids level, serum ALT level, serum AST level, and blood platelet count. Quartiles of n-3 PUFA (% TFA): Q1, <3.3; Q2, 3.3–4.2; Q3, 4.2–5.2; Q4, >5.2. Quartiles of n-6 PUFA (% TFA): Q1, <35.7; Q2, 35.7–38.4; Q3, 38.4–40.4; Q4, >40.4. Quartiles of MUFA (% TFA): Q1, <21.8; Q2, 21.8–23.5; Q3, 23.5–25.4; Q4, >25.4. Quartiles of SFA (% TFA): Q1, <32.7; Q2, 32.7–33.9; Q3, 33.9–35.2; Q4, >35.2. Source data are provided as a Source Data file.

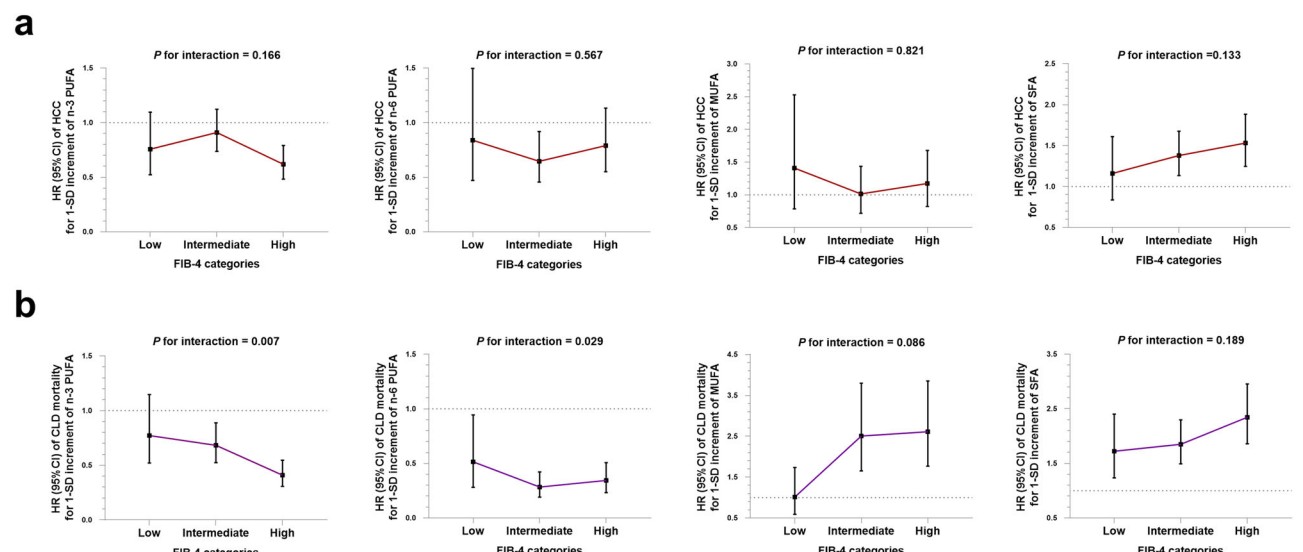

Fig. 3 | **Associations of specific fatty acids with incident HCC and CLD mortality by fibrosis stage (*n* = 252,398 participants).** **a** HRs of incident HCC associated with 1-SD increment in specific plasma fatty acids. **b** HRs of CLD mortality associated with 1-SD increment in specific plasma fatty acids. Multivariable Cox proportional hazard model was used. Model was adjusted for age, sex, ethnicity, BMI, waist circumference, Townsend deprivation index, education level, household

income, self-reported smoking status, self-reported frequency of alcohol intake, physical activity, diet quality score, baseline hypertension, baseline diabetes, baseline dyslipidemia, total cholesterol level, triglycerides level, total fatty acids level, serum ALT level, serum AST level, and blood platelet count. Data are presented as HRs and 95% CI. Source data are provided as a Source Data file.

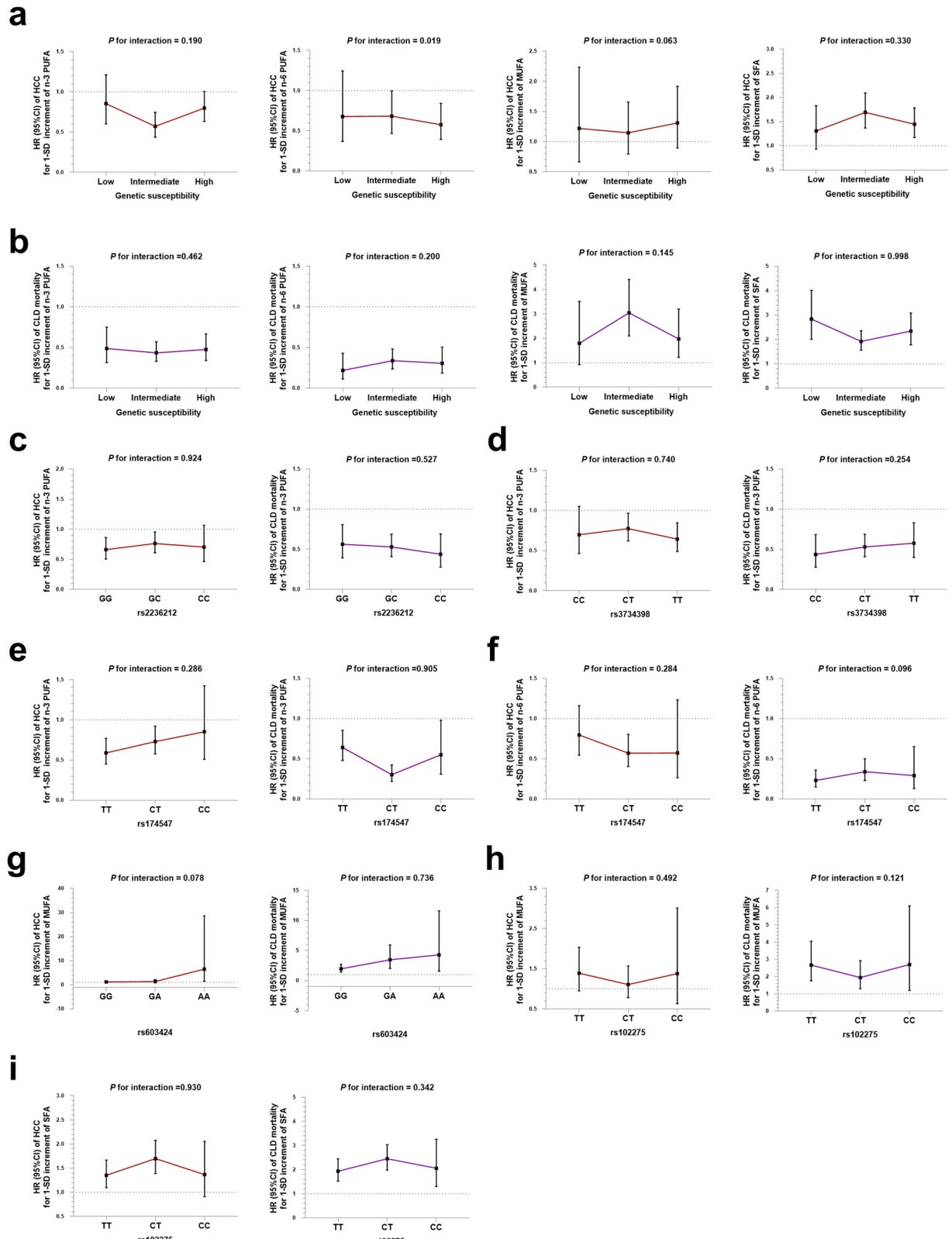

MUFA was only associated with a higher risk of CLD mortality. Besides, this study showed that acquired fibrosis risk significantly modified the associations of n-3 PUFA and n-6 PUFA with the risk of CLD mortality. Furthermore, PRS was found to interact with n-6 PUFA in the development of HCC. Our study contributes to the limited prospective evidence on the association between specific plasma fatty acids and end-stage liver outcomes.

To date, few prospective population-based studies have examined associations between plasma levels of specific fatty acids and severe liver outcomes. Our finding is similar to a Finnish cohort, in which PUFA (n-3 and n-6) were robustly inversely associated with fatty liver, but both MUFA and SFA were directly associated with fatty liver status[26]. The early stage of fatty liver is benign, yet this risk might translate to subsequent HCC. In a nested case-control study with 219

**Fig. 4 | Associations of specific fatty acids with incident HCC and CLD mortality by PRS or specific SNPs (*n* = 236,492 participants). a** HRs of incident HCC associated with 1-SD increment in specific plasma fatty acids according to PRS. **b** HRs of CLD mortality associated with 1-SD increment in specific plasma fatty acids according to PRS. **c** HRs of incident HCC and CLD mortality associated with 1-SD increment in plasma n-3 PUFA according to rs2236212 genotype. **d** HRs of incident HCC and CLD mortality associated with 1-SD increment in plasma n-3 PUFA according to rs3734398 genotype. **e** HRs of incident HCC and CLD mortality associated with 1-SD increment in plasma n-3 PUFA according to rs174547 genotype. **f** HRs of incident HCC and CLD mortality associated with 1-SD increment in plasma n-6 PUFA according to rs174547 genotype. **g** HRs of incident HCC and CLD mortality associated with 1-SD increment in plasma MUFA according to rs603424

genotype. **h** HRs of incident HCC and CLD mortality associated with 1-SD increment in plasma MUFA according to rs102275 genotype. **i** HRs of incident HCC and CLD mortality associated with 1-SD increment in plasma SFA according to rs102275 genotype Multivariable Cox proportional hazard model was used. Model was adjusted for age, sex, BMI, waist circumference, Townsend deprivation index, education level, household income, self-reported smoking status, self-reported frequency of alcohol intake, physical activity, diet quality score, baseline hypertension, baseline diabetes, baseline dyslipidemia, total cholesterol level, triglycerides level, total fatty acids level, serum ALT level, serum AST level, and blood platelet count. Data are presented as HRs and 95% CI. Source data are provided as a Source Data file.

liver cancer cases and 219 controls, MUFA were found to be positively associated with liver cancer risk while PUFA were inversely associated with risk[27]. In another study of biopsy-proven HCC patients, increased levels of SFA and MUFA but decreased levels of PUFA in plasma phospholipids were found in HCC patients relative to healthy controls as well as in cancerous tissue relative to its surrounding tissue[28]. Overall, our study expands the potential associations of specific fatty acids from the risk of HCC to a wide range of CLDs.

The adverse effects of SFA in the liver have been partly revealed. Correspondingly, we also showed strong and positive associations of SFA with HCC and CLD mortality. A single oral saturated fat load could rapidly increase hepatic lipid storage and insulin resistance, accompanied by regulation of hepatic gene expression and signaling that may contribute to the development of MASLD, the most prevalent CLD[29]. Furthermore, free SFA rapidly compounds with phospholipid species and is subsequently integrated into endoplasmic reticulum (ER) membranes, which can decrease the degree of fluidity of ER membranes and induce ER stress, an upstream signal in cellular dysfunction and apoptosis[30]. Mitochondria are crucial organelles responsible for cellular homeostasis. SFA could impair the efficiency of the respiratory transport chain, resulting in the production of reactive oxygen species and eventually leading to inflammation and apoptosis[31]. However, palmitic acid ($C_{16:0}$) is the initial product of de novo lipogenesis (DNL). Increased DNL was found in human HCC samples[32], and the extent of aberrant lipogenesis was correlated with clinical aggressiveness[33]. Altogether, it is suggested not only restricting SFA intake but also inhibiting abnormal DNL in the liver.

Dietary MUFA intake as well as the Mediterranean dietary pattern, characterized by a high ratio of plant-based MUFA to SFA, have been implied to be beneficial for liver health[34,35], which is inconsistent with our findings on circulating MUFA. One possible reason is that the endogenous synthesis of MUFA could not be ignored. Simon et al. reported that delta-9 desaturation impairment in the intestine leads to increased susceptibility to severe liver diseases such as NASH and HCC, indicating the importance of intestine-derived MUFA in homeostasis[36]. Another possible explanation is that hepatic stearoyl-CoA desaturase 1 (SCD-1) is the enzyme responsible for catalyzing the endogenous desaturation of SFA into MUFA and is activated in CLD[37]. Thus, elevated levels of MUFA may be considered a result of CLD. In addition, due to the induced DNL and increased storage of SFA in such conditions, the association of MUFA with liver diseases may be confounded by the adverse effect of SFA.

N-3 PUFA are considered to play an important role in the prevention and therapy of liver disease, which could decrease de novo lipogenesis, enhance mitochondrial fatty acid β-oxidation, and reduce hepatic inflammation[38,39]. Interestingly, we found that the protective association of PUFA with incident HCC was mainly driven by plasma non-DHA n-3 PUFA, which implied that the different PUFA components may have different functions. Regarding n-6 PUFA, it is controversial whether their effects are pro- or anti-inflammatory[40]. LA is the shortest-chained n-6 fatty acid, most of which is converted into arachidonic acid (AA), which is abundant in the membranes of cells involved in

inflammation. However, AA-derived lipoxins and cytochrome P450 derivatives were shown to have anti-inflammatory properties[41]. Notably, although both n-3 PUFA and n-6 PUFA showed negative associations with the outcomes in our study, their ratio (n-6/n-3) was positive, indicating that the balance between n-3 and n-6 PUFA is essential for liver homeostasis. The popularity of Western diets that contain excessive levels of n-6 PUFA but very low levels of n-3 PUFA is prone to lead to an unhealthy n-6/n-3 ratio of even 20:1[42]. It has been reported that AA and n-3 PUFA directly compete with one another for metabolism and that their mediators compete for receptors[43]. In addition, a high amount of dietary LA might limit endogenous eicosapentaenoic acid synthesis, potentially inducing a more inflammatory environment[44]. Therefore, our results support public health efforts that balance n-3 PUFA and n-6 PUFA intake to reduce the risk of chronic diseases[45].

In clinical practice, it is of significance to study the effect of exposure in different risk groups. We hereby analyzed whether there is a synergic effect of fatty acids and genetic or fibrosis risk on severe liver outcomes. Most CLDs can progress to liver fibrosis, which is a strong and independent predictor of mortality[46]. Hepatic stellate cells (HSCs) play a critical role in the progression of liver fibrosis. Zhang et al. discovered that n-3 PUFA downregulate the expression of profibrogenic genes in activated HSCs[47]. This finding may partly explain the interaction effect in our study. Besides, *PNPLA3*[I148M] minor allele carriers are associated with relative reductions in several n-6 PUFA[48], but how it influences HCC remains to be explored. Taken together, our findings demonstrated the importance of identifying individuals with higher fibrosis risk and measuring the levels of fatty acids.

The present study is the largest cohort study to date to evaluate the associations between NMR-based fatty acids and severe liver events. Despite the large sample size, long follow-up period, low loss to follow-up rate, and objective measurement of plasma fatty acids, some limitations need to be addressed. First, only baseline measurements of plasma fatty acids were made for each participant, rendering the data more prone to measurement error. However, it is reassuring that the concentrations of plasma fatty acids were moderately correlated with those measured at baseline in a subset of participants, indicating biological stability over time. Second, fatty acids were quantified by NMR from non-fasting plasma samples, which may bring greater variability than fasting samples. However, fasting duration only resulted in a small proportion of variation in plasma metabolic biomarker concentration[49]. Third, analyses of the relationships for more specific subtypes of fatty acids, such as different chain lengths and degrees of unsaturation of PUFA, are not allowed. In addition, hepatic free fatty acids, rather than esterized fatty acids, can serve as substrates for the formation of lipotoxic metabolites that cause liver injury[50], yet we cannot distinguish them here. Fourth, the study participants are of European ancestry and are individuals who are more health conscious[51], which potentially limits the generalizability of the results to other populations. However, the large sizes of exposure measures provide valid scientific inferences of associations between exposures and outcomes of interest. Fifth, although we performed multivariable-

adjusted models, the potential for residual confounding cannot be excluded. Due to the nature of observational research, a causal association cannot be demonstrated.

In conclusion, we showed that plasma SFA had adverse associations with incident HCC and CLD mortality, whereas plasma n-3 and n-6 PUFA had protective associations. Besides, fibrosis risk has an interaction effect on the observed associations. The present findings suggest that plasma fatty acids levels have important predictive value for severe liver outcomes.

# Methods

## Study populations
The UK Biobank received ethics approval from the North West Multi-Center Research Ethics Committee (REC No. 16/NW/0274), and all participants provided written electronic informed consent. This is a community-based cohort of 502,506 volunteers aged 39–73 years across England, Scotland and Wales. The baseline sociodemographic, lifestyle, and health-related data as well as blood samples were collected between March 2006 and October 2010.

The current project included 274,123 participants with valid data on NMR metabolic biomarkers. We then excluded those with end-stage liver diseases (Supplementary Table 1), unavailable genetic or FIB-4 data, and missing covariates, leaving 252,398 participants to analyze the associations. In addition, a subgroup of participants of European descent with available genetic data ($n = 236,492$) was extracted only for genetic analyses (Supplementary Fig. 1).

## Measurement of fatty acids
The measurements of metabolic biomarkers took place between June 2019 -April 2020 (phase 1) and April 2020 - June 2022 (phase 2) using a high-throughput nuclear magnetic resonance (NMR)-based profiling platform (Nightingale Health Ltd.). Altogether 251 metabolic biomarkers (including lipoprotein lipids in 14 subclasses, fatty acids and fatty acid compositions, as well as various low-molecular weight metabolites) for EDTA plasma samples (collected at baseline) were measured from a randomly selected subset of approximately 280,000 UK Biobank participants. Details of the Nightingale Health NMR biomarker platform and experiment process have been described previously[52].

The primary exposure was specific plasma fatty acids [n-3 PUFA], n-6 PUFA, MUFA, SFA, LA ($C_{18:2}$n-6, one of the essential fatty acids), and DHA ($C_{22:6}$n-3), expressed as percentages of total plasma fatty acids (% TFA). During the first repeat assessment visit (2012–2013), 14,609 participants provided plasma samples for NMR analysis again, which demonstrated modest correlations with baseline levels [$r = 0.47$ (SFA)−0.66 (MUFA), Supplementary Fig. 2]. The quantified plasma fatty acids represent a combination of free fatty acids as well as fatty acids in lipid fractions such as triglycerides, phospholipids, or cholesterol esters.

## Dietary assessment
At baseline, each participant was asked to complete a brief touchscreen food frequency questionnaire covering the types and frequency of consumption of food groups and drinks over the past year. Then, we created a diet quality score based on the following 10 foods: vegetables, fruits, fish, dairy, whole grains, vegetable oils, refined grains, processed meats, unprocessed red meats, and sugar-sweetened beverages (Supplementary Table 2). Each dietary component was scored from 0 (unhealthiest) to 10 (healthiest) points, and a higher score represents a higher intake of vegetables, fruits, fish, dairy, whole grains, and vegetable oils or a lower intake of refined grains, processed meats, unprocessed red meats, and sugar-sweetened beverages. The total diet quality score was the sum of all the diet component scores, with a higher score representing a higher overall diet quality[53].

## Outcome ascertainment
All participants gave consent for continued follow-up by linkage to electronic health records, including hospital records (held by the Department of Health's Hospital Episode Statistics and the Scottish Morbidity Record), and death records as well as cancer registers (maintained by the Office for National Statistics and the Registrar General's Office). Each record was coded using the International Classification of Diseases, 10th revision (ICD-10).

We examined 2 primary endpoints: (i) incident cases of HCC (ICD-10 code C22.0) and (ii) fatal (underlying cause) CLD, defined as death from alcoholic liver diseases, liver fibrosis, cirrhosis, and MASLD (ICD-10 codes K70, K74, K75.8, and K76.0).

At the time of analysis, the updating dates of linkages to hospital inpatient admission, cancer registries and death registries were 31 October 2022, 1 June 2022 and 19 December 2022, respectively. Follow-up time in person-years was calculated from the date of attendance until the date of HCC diagnosis or death, whichever occurred earlier.

## Genetic analyses
Genotyping in the UK Biobank was performed on two closely related purpose-designed arrays: the UK BiLEVE and UK Biobank Axiom[54]. We selected 5 single-nucleotide polymorphisms (SNPs, Supplementary Table 3): rs738409 [patatin-like phospholipase domain-containing protein 3 (PNPLA3) I148M variant], rs58542926 [transmembrane 6 superfamily member 2 (TM6SF2) E167K], rs641738 [C > T membrane bound O-acyltransferase domain containing 7 (MBOAT7)], rs1260326 ([glucokinase regulator] (GCKR) P446L) and rs72613567 [17β-hydroxysteroid dehydrogenase type 13 (HSD17B13):TA], which are associated with cirrhosis and HCC[55]. Then, according to the number of risk alleles of each SNP, we calculated the polygenic risk scores (PRS) as below:

$$PRS = 0.266 \times PNPLA3 + 0.274 \times TM6SF2 + 0.065 \times GCKR + 0.063 \times MBOAT7 - 0.361 \times HSD17B13 \tag{1}$$

We defined genetic risk as follows: "low risk" (lowest quartile of PRS), "intermediate risk" (second to third quartiles of PRS), and "high risk" (highest quartile of PRS).

We also selected SNPs associated with specific circulating levels of fatty acids at the genome-wide significance (Supplementary Table 4)[20–22].

## Fibrosis evaluation
The evaluation of liver fibrosis status was calculated by the fibrosis 4 (FIB-4) score, which is a commonly used non-invasive fibrosis marker[56]:

$$FIB - 4 = \frac{Age\,(years) \times AST\,(U/L)}{Platelet\,count\,(10^9/L) \times \sqrt{ALT(U/L)}} \tag{2}$$

We categorized participants into low-, intermediate- and high-risk groups for advanced fibrosis according to the following suggested cutoffs: <1.30 (low risk), 1.30–2.67 (intermediate risk) and >2.67 (high risk).

## Covariates
During the baseline visit, information on age, sex, ethnicity, Townsend deprivation index (a measure of socioeconomic status), education levels, household income, smoking status and alcohol intake frequency was collected by nurse-led interviews and self-administered touchscreen questionnaires. Body size measurements (weight, height, and waist circumference) were obtained by trained staffs using standardized procedures and regularly calibrated equipment. Body mass index (BMI) was calculated using the standard formula [weight (kilograms)/height (meters) squared]. Physical activity was evaluated by the metabolic equivalent task (MET) minutes based on the

International Physical Activity Questionnaire (IPAQ). Hypertension was defined as systolic pressure ≥140 mmHg, diastolic pressure ≥90 mmHg, use of medications for blood pressure or self-reported or diagnosed by a doctor. Diabetes was defined as blood glucose ≥11.1 mmol/L, glycated hemoglobin (HbA1c) ≥ 48 mmol/mol, use of insulin or self-reported or diagnosed by a doctor. Dyslipidemia was defined as self-reported history of high cholesterol, hospital diagnosis of hyperlipidemia (ICD-10 E78.0, E78.1, E78.2, E78.4, E78.5) or use of lipid-lowering drugs. In addition, plasma levels of total cholesterol, triglycerides, and total fatty acids were also measured on the NMR platform simultaneously. Details of these assessments are available from https://biobank.ctsu.ox.ac.uk/crystal/browse.cgi.

### Statistical analyses

Plasma levels of specific fatty acids were expressed as percentages of plasma total fatty acids[57], each of which was then divided into quartiles to compare baseline characteristics or risk for incident HCC and CLD mortality across them. To describe participant baseline characteristics for each quartile, continuous variables were presented as the mean ± standard deviation (SD) and categorical variables were presented as percentages. Cox proportional hazards regression models were used to estimate the risk for incident HCC and CLD mortality. Hazard ratios (HRs) and 95% confidence intervals (CIs) for each quartile of exposure were reported, with the lowest quartile as the reference. For each outcome, 3 separate models with increasing adjustment were run: model 1 was adjusted for age, sex, and ethnicity (White, Asian, Black, or other ethnic groups). Model 2 was adjusted for model 1 plus BMI, waist circumference, Townsend deprivation index (quintiles), education level (university/college degree or others), household income (less than £18,000, £18,000 to £30,999, £31,000 to £51,999, £52,000 to £100,000, greater than £100,000, or do not know/prefer not to answer), self-reported smoking status (never, former or current smoker), self-reported frequency of alcohol intake (daily/almost daily, 1–4 times a week, 1–3 times a month, or special occasions only/never), physical activity [<600 (inadequate), 600–3000 (moderate), >3000 (vigorous) metabolic equivalent of energy (MET) minutes per week, or missing], diet quality score, baseline hypertension (yes/no), baseline diabetes (yes/no), and baseline dyslipidemia (yes/no). Model 3 was adjusted for model 2 plus total cholesterol level (measured by NMR), triglycerides level (measured by NMR), total fatty acids level, serum ALT level, serum AST level, and blood platelet count.

To investigate whether fibrosis stage or genetic predisposition would modify the associations between fatty acids and liver-related events, we included an interaction term in the regression model. The HR of the product term was the measure of interaction, and the Wald test was utilized to evaluate whether this term was statistically significant.

We performed subgroup analyses stratified by baseline characteristics to test the potential modification effect of covariates on the associations of plasma fatty acids with HCC and CLD mortality. In sensitivity analyses, we further excluded liver-related events in the first two years of follow-up to minimize the possibility of reverse causation. We further adjusted for the history of lipid-lowering medication (Atorvastatin, Crestor, Eptastatin, Fluvastatin, Lescol, Lipitor, Lipostat, Pravastatin, Rosuvastatin, Simvador, Simvastatin, or Zocor), PRS, *FADS1/2* genotype, and remaining plasma fatty acids (n-3 PUFA, n-6 PUFA, MUFA, SFA) where appropriate.

All analyses were conducted using SAS version 9.4. The *P* values for all statistical analyses were two-sided, with a significance level of 0.05.

### Reporting summary

Further information on research design is available in the Nature Portfolio Reporting Summary linked to this article.

## Data availability

The datasets analyzed during the current study are available in a public, open access repository (https://www.ukbiobank.ac.uk/). The UK Biobank data are available for approved researchers through the UK Biobank data-access protocol. This research has been conducted using the UK Biobank resource under application number 79302. Source data are provided with this paper.

## Code availability

SAS codes for the analyses are available at https://github.com/liuznCHN/UKB.

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

## Acknowledgements

We thank the participants of the UK Biobank. This work was supported by the National Natural Science Foundation of China (82370574[Xu], 82070585[Xu], and 82300654[Xie]).

## Author contributions

Study concept and design: Z.L, C.X; Analysis and interpretation of the data: Z.L, H.H, J.X; Drafting of the manuscript: Z.L, H.H, J.X, Y.X, C.X; Critical revision of the manuscript: Z.L, C.X.

## Competing interests

The authors declare no competing interests.
