## [Peer Review File · Nature Communications]

Circulating fatty acids and risk of hepatocellular carcinoma and chronic liver disease mortality in the UK BiobankREVIEWER COMMENTS

Reviewer #1 (Remarks to the Author):

The manuscript "Association of circulating fatty acids with risk of hepatocellular carcinoma and chronic liver disease mortality: evidence from the UK Biobank cohort study" by Liu et al. uses ~65,000 participants from the UK Biobank with NMR profiling using the Nightingale platform to assess the relationship between circulating fatty acids risk of both hepatocellular carcinoma (HCC) and chronic liver disease (CLD) mortality. Cox proportional hazards models were used to assess incident cases and mortality in three models that include a minimally to fully adjusted models. Furthermore, the authors adjust/assess the relationship of underlying fibrosis risk and genetics by including them as interaction terms.

The clear strength of this study is the number of individuals that were used, which maximizes statistical power. This is very important when outcomes are rare, as is this case for HCC and CLD mortality. However, there are several points of clarification, analysis, and potential confounding that need to be addressed in order to more robustly evaluate this association. Prior studies have identified findings with FA subclasses and HCC/CLD, but there are some inconsistencies among these findings. This study is well-powered to clarify these associations; however, given the stark differences in the distributions and many covariates and levels of each FA subclass, there are concerns with residual confounding.

My specific points follow:

- 1) The authors categorize fatty acids into saturated, monosaturated, polyunsaturated (n-3 and n-6 together). Given that the Nightingale platform has absolute quantification and that several of these compounds may also be best understood in ratios, it the ratios to total FA should also be provided for these association tests. This seems critically important, as the ratios can often be more important than each metabolite class separately.
- 2) Since both linoleic acid and docosahexaenoic acid measures are provided, these should also be included given their important role in this pathway. While these are in the supplement, there is no mention of them at all in the paper.
- 3) It is important to show the correlation of these metabolite groups with each other (n-3, n-6 PUFAs/MFAs/SFAs). This may be important in understanding the overall relationship.
- 4) Since FFQ data are also available on these individuals, it would be interesting and important to see whether the circulating levels correlate with reported food intake in the FFQ
- 5) A major reason to show Table 1 stratified by exposure is to evaluate whether other characteristics vary by the exposure. If they do, then these are likely important covariates/confounders to include in the analysis. It is important to demonstrate that the analysis controls for potential confounding between the exposures and the outcomes. In the analyses presented, it is very challenging given that the group of individuals with higher n-3 PUFAs are distinctly different from those with lower n-3 levels for almost every characteristic presented in Table 1. The authors should identify which ones of these also vary by outcome to identify confounders and then adjust for all of these. The authors address for several covariates using the multiple models; however, there is still concern of residual confounding. Table 1 should include therefore include a larger list of variables, including several comorbidities and metrics for diet quality. A more complete fully adjusted model should be included that has all variables that are associated with the outcomes and exposures. If you wanted to show how multiples models relate with regard to significance and HRs, this can also be done using a heatmap, which may summarize the findings in a visual, yet more absorbable way (with actual metrics included in a supplemental table).
- 6) Given how all of the different covariates are different by the exposure, this really puts into question the potential of residual confounding. This begs the question: If the individuals in the high and lower quartiles were the same on other factors would the observed association between high/low n-3 PUFAs still be observed? Given that this study is very large, stratified analyses of comparable groups is possible and are an excellent way to address this question. It would be

important to demonstrate that the observed associations remain when you stratify of these major confounders, including BMI, education/income, non-smokers (assuming that there is sufficient power to do this).

7) It is unclear why only 2 SNPs were adjusted for in the adjustment for genetics. Unless these outcomes have Mendelian transmission or are driven by primarily these variants, this is an insufficient approach for assessing underlying genetic risk. Why weren't something like polygenic risk scores used given that there are publications to compute these(<https://doi.org/10.1016/j.jhep.2020.11.024>)? Moreover, why are those two SNPs selected more specifically. Especially since this pathway is dependent of major met-QTLs, including FADS1/2, it is curious why only those two SNPs were selected. It is also not clear that these are equally important for HCC and CLD. A more robust adjustment for underlying genetics is necessary. Finally, there is not reporting of the main genetic effects for these findings. It would be important to report both main effects and interaction effects for the overall interpretation (if using a PRS, then apply this approach with the PRS).

8) It is unclear with the fibrosis risk score is and the rationale for using it. If this is going to be published in a more general journal, a few sentences of background is helpful. This needs to be explained more robustly. Moreover, the main effects of these categories should also be presented for the outcome, as well as the interaction terms.

9) The sensitivity analyses are more enlightening when all conditions are done together in one model. This should be performed to see how all things together affect the results.

10) Supplemental figures S3 and S4 are more important than showing the multiple models. Please only show the fully adjusted model as well as these plots. It would also be more effective to show the main results of the ORs in a figure with plots of the HRs with SEs in the main paper.

11) If you stratify the associations for CLD by HCC, NAFLD, AFLD, and viral hepatitis, are the same associations observed for all pre-disposing conditions of CLD (I recognize that power may be limited if this is done, so it might it be possible)?

12) It is notable that non-white participants are removed. What is the justification for this? There are >14,000 non-white individuals removed. Even if the genetic analyses are not possible (which I still think that they are likely possible), it would seem very important to include these in the other analyses, especially since there are a large number of minority groups and there are likely not many other places to study them with regard to these outcomes and FA subclasses. In addition, this adds another point of novelty to this study.

Reviewer #2 (Remarks to the Author):

ABSTRACT

Line 52: Redundant of "for MUFAs". Please remove.

INTRODUCTION

Since there are several prospective cohort studies examined the association between fatty acids and HCC, the authors should have laid out a strong rationale for the current study, such as using biomarker based (thought they did mention somewhere, yet it did not clear to me).

Another issue is that they need to justify why they chose 2 endpoints: incident HCC and LCD mortality for the current analysis.

Please provide citation(s) for paragraph 2, lines 75-81.

Line 93: spell out TG the first-time use.

Line 102: The NAFLD has new nomenclature, MASLD. Please update this.

DISCUSSION

Line 197: since the analysis in the Finnish cohort study included both cross-sectional and prospective type, your statement in prior lines (195-196) is incorrect. Please modify it.

METHODS

Statistical analysis: (lines 380-382)

Please describe more details why you chose quartile scale and how you did it.

Supplementary Tables S4 and S5. Footnote

- Please list all covariates that were adjusted in the "fully adjusted model"

RESPONSE TO REVIEWERS' COMMENTS

Reply to Reviewer #1:

GENERAL COMMENT

The manuscript “Association of circulating fatty acids with risk of hepatocellular carcinoma and chronic liver disease mortality: evidence from the UK Biobank cohort study” by Liu et al. uses ~65,000 participants from the UK Biobank with NMR profiling using the Nightingale platform to assess the relationship between circulating fatty acids risk of both hepatocellular carcinoma (HCC) and chronic liver disease (CLD) mortality. Cox proportional hazards models were used to assess incident cases and mortality in three models that include a minimally to fully adjusted models. Furthermore, the authors adjust/assess the relationship of underlying fibrosis risk and genetics by including them as interaction terms.

The clear strength of this study is the number of individuals that were used, which maximizes statistical power. This is very important when outcomes are rare, as is this case for HCC and CLD mortality. However, there are several points of clarification, analysis, and potential confounding that need to be addressed in order to more robustly evaluate this association. Prior studies have identified findings with FA subclasses and HCC/CLD, but there are some inconsistencies among these findings. This study is well-powered to clarify these associations; however, given the stark differences in the distributions and many covariates and levels of each FA subclass, there are concerns with residual confounding.

Reply: Thank you very much for your valuable comments and suggestions. We revised the manuscript accordingly, and point-by-point replies to the comments are shown below.

Comment 1: *The authors categorize fatty acids into saturated, monosaturated, polyunsaturated (n-3 and n-6 together). Given that the Nightingale platform has absolute quantification and that several of these compounds may also be best understood in ratios, it the ratios to total FA should also be provided for these association tests. This seems critically important, as the ratios can often be more*

important than each metabolite class separately.

Reply: Thank you very much for this suggestion. It is true that specific plasma FA levels expressed as percentages of plasma total FA are more meaningful than absolute concentrations in terms of metabolic relationships. Correspondingly, we expressed the specific plasma fatty acids (n-3 PUFAs, n-6 PUFAs, MUFAs, SFAs, linoleic acid (LA), and docosahexaenoic acid (DHA) as percentages of total plasma fatty acids (% TFA) in our study. Besides, we explored the ratio of n-6 PUFAs to n-3 PUFAs (n-6/n-3), and the ratio of MUFA to SFA (MUFA/SFA) in our revised manuscript. We have added these data in the revised manuscript on Lines 364-366 and in revised Figure 1, Figure 2, Tables S6, S7 and S9.

Lines 364-366: **The primary exposure was specific plasma fatty acids [n-3 PUFA, n-6 PUFA, MUFA, SFA, LA (C_{18:2}n-6, one of the essential fatty acids), and DHA (C_{22:6}n-3), expressed as percentages of total plasma fatty acids (% TFA).**

Figure legend 1 and Figure legend 2:

Quartiles of n-3 PUFA (% TFA): Q1, <3.3; Q2, 3.3–4.2; Q3, 4.2–5.2; Q4, >5.2.

Quartiles of n-6 PUFA (% TFA): Q1, <35.7; Q2, 35.7–38.4; Q3, 38.4–40.4; Q4, >40.4.

Quartiles of MUFA (% TFA): Q1, <21.8; Q2, 21.8–23.5; Q3, 23.5–25.4; Q4, >25.4.

Quartiles of SFA (% TFA): Q1, <32.7; Q2, 32.7–33.9; Q3, 33.9–35.2; Q4, >35.2.

Supplementary Table S6. Associations between plasma n-3 PUFA, n-6 PUFA, MUFA, SFA levels and incident HCC risk (part)

Supplementary Table S8. Associations between plasma n-3 PUFA, n-6 PUFA, MUFA, SFA levels and CLD mortality (part)

Fatty acids	Range (% TFA)
n-3 PUFA	
Q1	<3.3
Q2	3.3–4.2
Q3	4.2–5.2
Q4	>5.2
n-6 PUFA	
Q1	<35.7

Q2	35.7–38.4
Q3	38.4–40.4
Q4	>40.4
MUFA	
Q1	<21.8
Q2	21.8–23.5
Q3	23.5–25.4
Q4	>25.4
SFA	
Q1	<32.7
Q2	32.7–33.9
Q3	33.9–35.2
Q4	>35.2

Supplementary Table S7. Associations between plasma LA, DHA, n-6/n-3, MUFA/SFA and incident HCC risk (part)

Supplementary Table S9. Associations between plasma LA, DHA, n-6/n-3, MUFA/SFA and CLD mortality (part)

	Range†
LA (n-6)	
Q1	<25.2
Q2	25.2–27.5
Q3	27.5–29.7
Q4	>29.7
DHA (n-3)	
Q1	<1.3
Q2	1.3–1.7
Q3	1.7–2.0
Q4	>2.0
n-6/n-3	
Q1	<7.2
Q2	7.2–9.0
Q3	9.0–11.4
Q4	>11.4
MUFA/SFA	
Q1	<0.65
Q2	0.65–0.69
Q3	0.69–0.75
Q4	>0.75

† The unit of LA and DHA were %TFA.

Comment 2: Since both linoleic acid and docosahexaenoic acid measures are provided,

these should also be included given their important role in this pathway. While these are in the supplement, there is no mention of them at all in the paper.

Reply: Thank you for this suggestion. We have explained these data in the revised manuscript on Lines 155-160 and Lines 172-176.

Lines 155-160: Additionally, linoleic acid (LA) showed no significant association with the risk of HCC ($P_{\text{trend}}=0.617$), and docosahexaenoic acid (DHA) only showed a negative tendency ($P_{\text{trend}}=0.042$) in the fully adjusted model. N-6/n-3 showed a positive association with HCC [HR_{Q4vsQ1}: 1.95 (1.35–2.82), $P_{\text{trend}}<0.001$], but MUFA/SFA showed a negative association [HR_{Q4vsQ1}: 0.48 (0.31–0.76), $P_{\text{trend}}=0.008$, Supplementary Table S7].

Lines 172-176: Correspondingly, LA and DHA were also significantly associated with a lower risk of CLD mortality [HR_{Q4vsQ1}: 0.35 (0.20–0.63) and 0.24 (0.14–0.39)]. N-6/n-3 showed a positive association [HR_{Q4vsQ1}: 3.61 (2.39–5.45), $P_{\text{trend}}<0.001$], but MUFA/SFA showed a negative association [HR_{Q4vsQ1}: 0.49 (0.30–0.81), $P_{\text{trend}}<0.001$, Supplementary Table S9].

Comment 3: *It is important to show the correlation of these metabolite groups with each other (n-3, n-6 PUFAs/MFAs/SFAs). This may be important in understanding the overall relationship.*

Reply: Thank you for this comment. We have analyzed the correlation of these metabolite groups with each other in the revised manuscript, and presented the data on Lines 127-132 and in Supplementary Table S4.

Lines 127-132: The Spearman correlations between specific plasma fatty acids are summarized in Supplementary Table S4. Plasma levels of n-3 PUFA had weak negative correlations with n-6 PUFA ($r_s = -0.137$), MUFA ($r_s = -0.278$) and SFA ($r_s = -0.137$). Plasma levels of n-6 PUFA had strong negative correlations with MUFA ($r_s = -0.750$) and SFA ($r_s = -0.646$). There was a weak positive correlation between plasma SFA and MUFA ($r_s = 0.265$)

Supplementary Table S4. Spearman correlations between specific plasma fatty

acids (% of total fatty acids).

	n-3 PUFA	n-6 PUFA	MUFA	SFA
n-3 PUFA	1.000	-0.137	-0.278	-0.137
n-6 PUFA		1.000	-0.750	-0.646
MUFA			1.000	0.265
SFA				1.000

All with $P < 0.001$

Comment 4: Since FFQ data are also available on these individuals, it would be interesting and important to see whether the circulating levels correlate with reported food intake in the FFQ.

Reply: Thank you very much for this valuable comment and suggestion. Each participant was asked to complete a brief touchscreen food frequency questionnaire (FFQ) covering the types and the frequency of consumption of food groups and drinks over the past year. Accordingly, we created a diet quality score based on 10 foods: vegetables, fruits, fish, dairy, whole grains, vegetable oils, refined grains, processed meats, unprocessed red meats, and sugar-sweetened beverage. Each dietary component was scored from 0 (unhealthiest) to 10 (healthiest) points, and the total diet quality score was the sum of all the diet component scores and ranged from 0 to 100, with a higher score representing a higher overall diet quality (*Diabetes Care*, 2021, 44(11), 2470–2479; *Nutrients*, 2023, 15(2), 271).

In the revised manuscript, we reported the correlation of each dietary component with specific FAs in Supplementary Figure S3. We found that individuals with higher plasma n-3 PUFA levels consumed higher fruits, vegetables, whole grains, fish, but lower refined grains and processed meats. On the contrary, those with higher plasma SFA levels consumed less fruits, vegetables, whole grains, vegetable oils, but more refined grains, processed meats, and unprocessed red meats. We have added these data to the revised manuscript as follows.

Lines 122-126: **As for detailed dietary factors (Supplementary Figure S3), individuals with higher plasma n-3 PUFA levels consumed higher fruits, vegetables, whole grains, fish, but lower refined grains and processed meats. On the contrary, those**

with higher plasma SFA levels consumed less fruits, vegetables, whole grains, vegetable oils, but more refined grains, processed meats, and unprocessed red meats.

Lines 374-384: At baseline, each participant was asked to complete a brief touchscreen food frequency questionnaire covering the types and the frequency of consumption of food groups and drinks over the past year. Then, we created a diet quality score based on the following 10 foods: vegetables, fruits, fish, dairy, whole grains, vegetable oils, refined grains, processed meats, unprocessed red meats, and sugar-sweetened beverage (Supplementary Table S2). Each dietary component was scored from 0 (unhealthiest) to 10 (healthiest) points, and a higher score represents higher intake of vegetables, fruits, fish, dairy, whole grains, and vegetable oils or lower intake of refined grains, processed meats, unprocessed red meats, and sugar-sweetened beverages. The total diet quality score was the sum of all the diet component scores, with a higher score representing a higher overall diet quality.

Supplementary Figure S3. Scores of individual dietary components for specific fatty acids levels

Comment 5: A major reason to show Table 1 stratified by exposure is to evaluate whether other characteristics vary by the exposure. If they do, then these are likely important covariates/confounders to include in the analysis. It is important to demonstrate that the analysis controls for potential confounding between the exposures and the outcomes. In the analyses presented, it is very challenging given that the group of individuals with higher n-3 PUFAs are distinctly different from those with lower n-3 levels for almost every characteristic presented in Table 1. The authors should identify which ones of these also vary by outcome to identify confounders and then adjust for all of these. The authors address for several covariates using the multiple models; however, there is still concern of residual confounding. Table 1 should include therefore include a larger list of variables, including several comorbidities and metrics for diet quality. A more complete fully adjusted model should be included that has all variables that are associated with the outcomes and exposures. If you wanted to show how multiples models relate with regard to significance and HRs, this can also be done using a heatmap, which may summarize the findings in a visual, yet more absorbable way (with actual metrics included in a supplemental table).

Reply: Thank you for this valuable suggestion. Comorbidities (hypertension, diabetes, and dyslipidemia) and diet quality are included in Table 1 in the revised manuscript.

Besides, we rebuilt a more complete fully adjusted model adjusting for other potential covariates/confounders. Those included age, sex, ethnicity (White, Asian, Black, or other ethnic groups), BMI, waist circumference, Townsend deprivation index (quintiles), education level (university/college degree or others), household income (less than £18,000, £18,000 to £30,999, £31,000 to £51,999, £52,000 to £100,000, greater than £100,000, or do not know/prefer not to answer), self-reported smoking status (never, former or current smoker), self-reported frequency of alcohol intake (daily/almost daily, 1–4 times a week, 1–3 times a month, or special occasions only/never), physical activity [<600 (inadequate), 600–3000 (moderate), >3000 (vigorous) metabolic equivalent of energy (MET) minutes per week, or missing], diet quality score, baseline hypertension (yes/no), baseline diabetes (yes/no), baseline

dyslipidemia (yes/no), total cholesterol level (measured by NMR), triglycerides level (measured by NMR), total FA level, serum ALT level, serum AST level, and blood platelet level.

In addition, we used heatmaps (Figure 1 and Figure 2) to summarize the findings as suggested, with actual metrics included in Supplementary Table S6 and S8. We have added these data in the revised manuscript as follows.

Lines 458-472: For each outcome, 3 separate models with increasing adjustment were run: model 1 was adjusted for age, sex, and ethnicity (White, Asian, Black, or other ethnic groups). Model 2 was adjusted for model 1 plus BMI, waist circumference, Townsend deprivation index (quintiles), education level (university/college degree or others), household income (less than £18,000, £18,000 to £30,999, £31,000 to £51,999, £52,000 to £100,000, greater than £100,000, or do not know/prefer not to answer), self-reported smoking status (never, former or current smoker), self-reported frequency of alcohol intake (daily/almost daily, 1–4 times a week, 1–3 times a month, or special occasions only/never), physical activity [<600 (inadequate), 600–3000 (moderate), >3000 (vigorous) metabolic equivalent of energy (MET) minutes per week, or missing], diet quality score, baseline hypertension (yes/no), baseline diabetes (yes/no), baseline dyslipidemia (yes/no). Model 3 was adjusted for model 2 plus total cholesterol level (measured by NMR), triglycerides level (measured by NMR), total FA level, serum ALT level, serum AST level, and blood platelet count.

Table 1. Population summary characteristics (part)

Variables
Male (%)
Age (years)
Ethnicity (%)
White
Asian
Black
Others

Townsend deprivation index
College or university degree (%)
Household income (£)
 <18,000
 18,000 to 30,999
 31,000 to 51,999
 52,000 to 100,000
 >100,000
Physical activity (%)
 Inadequate
 Moderate
 Vigorous
Smoking status (%)
 Never
 Previous
 Current
Alcohol consumption (%)
 Never or special occasions only
 1 to 3 times/month
 1 to 4 times/week
 Daily or almost daily
Waist circumference (cm)
Body mass index (kg/m²)
Hypertension (%)
Diabetes (%)
Dyslipidemia (%)
Platelet (10⁹/L)
Alanine aminotransferase (U/L)
Aspartate aminotransferase (U/L)
Total cholesterol (mmol/L)
Triglycerides (mmol/L)
Total fatty acid (mmol/L)
Diet quality score

Figure 1. Associations between plasma n-3 PUFA, n-6 PUFA, MUFA, SFA levels and incident HCC risk

Figure 2. Associations between plasma n-3 PUFA, n-6 PUFA, MUFA, SFA levels and risk of CLD mortality

Comment 6: Given how all of the different covariates are different by the exposure, this really puts into question the potential of residual confounding. This begs the question: If the individuals in the high and lower quartiles were the same on other factors would the observed association between high/low n-3 PUFAs still be observed? Given that

this study is very large, stratified analyses of comparable groups is possible and are an excellent way to address this question. It would be important to demonstrate that the observed associations remain when you stratify of these major confounders, including BMI, education/income, non-smokers (assuming that there is sufficient power to do this).

Reply: Thank you for this valuable comment and suggestion. We performed subgroup analyses stratified by baseline characteristics to test the potential modification effect of covariates on the associations of plasma fatty acids with HCC and CLD mortality. The results were shown on Lines 213-228 and in Supplementary Table S13-S16.

Lines 213-228: In subgroup analyses, the negative associations of plasma n-3 PUFA with HCC risk (Supplementary Table S13) were more pronounced among those less than 60 years ($P_{\text{interaction}}=0.034$), and the association of plasma n-3 PUFA with CLD mortality were more pronounced among those over 60 years ($P_{\text{interaction}}=0.041$) and less social deprived ($P_{\text{interaction}}=0.010$). For n-6 PUFA (Supplementary Table S14), the negative associations with HCC risk were more pronounced among those more than 60 years ($P_{\text{interaction}}=0.003$), and the negative associations with CLD mortality were more pronounced among men ($P_{\text{interaction}}=0.001$), those with BMI less than 25 kg/m² ($P_{\text{interaction}}<0.001$), and those drank more frequently ($P_{\text{interaction}}<0.001$). In addition, we observed stronger positive associations of plasma MUFA with HCC risk among older people ($P_{\text{interaction}}=0.003$), and stronger positive associations of MUFA with CLD mortality among men ($P_{\text{interaction}}=0.006$), those with BMI less than 25 kg/m² ($P_{\text{interaction}}=0.020$), those drank more frequently ($P_{\text{interaction}}=0.002$), and those without a history of diabetes ($P_{\text{interaction}}=0.044$, Supplementary Table S15). Additionally, the positive associations of plasma SFA and CLD mortality were more evident among men ($P_{\text{interaction}}=0.002$, Supplementary Table S16).

Comment 7: *It is unclear why only 2 SNPs were adjusted for in the adjustment for genetics. Unless these outcomes have Mendelian transmission or are driven by primarily these variants, this is an insufficient approach for assessing underlying genetic risk. Why weren't something like polygenic risk scores used given that there*

are publications to compute these (<https://doi.org/10.1016/j.jhep.2020.11.024>)? Moreover, why are those two SNPs selected more specifically. Especially since this pathway is dependent of major met-QTLs, including FADS1/2, it is curious why only those two SNPs were selected. It is also not clear that these are equally important for HCC and CLD. A more robust adjustment for underlying genetics is necessary. Finally, there is not reporting of the main genetic effects for these findings. It would be important to report both main effects and interaction effects for the overall interpretation (if using a PRS, then apply this approach with the PRS).

Reply: Thank you for this suggestion. It is true that integrating the number and effect sizes of these risk alleles to construct a PRS can continuously and quantitatively depict the genetic risk of diseases. Thus, we calculated the PRS as you suggested instead of using single SNPs (rs738409 and rs58542926).

In addition, it has been shown that the levels of circulating fatty acids are partly under genetic regulation. We concentrated our efforts on SNPs that are available in the UK biobank and located in or close to genes (*FADS1/2*, *SCD* and *ELOVL2*) encoding enzymes with a central role in the metabolic pathway of fatty acids (*PLOS Genetics*, 2011;7: e1002193; *Circ Cardiovasc Genet*, 2013;6:171-183; *Circ Cardiovasc Genet*, 2014;7:321-331), including rs174547 (*FADS1*), rs2236212 (*ELOVL2*), rs3734398 (*ELOVL2*), rs603424 (*SCD/PKD2L1*), and rs102275 (*FADS1/2*). Detailed information is listed below (Supplementary Table S4).

Overall, evaluating the interaction of circulating biomarkers of FAs with genetics on the risk of liver disease could help develop personalized advice based on the genotype. Discovering the interaction of genetics with circulating FAs could also be informative for preventative approaches.

In addition, we further adjusted for PRS or FADS1/2 genotype in the sensitivity analyses (Supplementary Tables S11 and S12).

We also reported both main effects and interaction effects in Supplementary Figure S7 and Figure 4.

Lines 92-97: **In addition to dietary consumption, the levels of circulating fatty acids also depend on endogenous synthesis and metabolism (20, 21). In individuals of**

European descent, the polymorphism of fatty acid desaturases (*FADS1/2* and *SCD*) and fatty acid elongases (*ELOVL2*) have been found to influence the levels of MUFA, SFA (20), as well as PUFA (21, 22). However, it is unknown whether these variants could modify the effects of specific fatty acids on the liver related outcomes.

Lines 194-205: For PRS, higher genetic susceptibility was associated with a higher risk of HCC and CLD mortality ($P_{trend} < 0.001$ and $P_{trend} = 0.023$, respectively. Supplementary Figure S7a). Significant interaction of plasma n-6 PUFA with PRS on HCC risk ($P_{interaction} = 0.019$) were detected. In category analysis, the inverse associations of plasma n-6 PUFA with HCC were stronger among participants with higher PRS [HR_{1-SD} (95% CI): 0.58 (0.39–0.84), Figure 4a]. However, no significant interactions were observed between the other plasma fatty acids and PRS for HCC and CLD mortality (Figure 4a-4b). In addition, neither the main effects of rs2236212, rs3734398, rs174547, rs603424, and rs102275 genotypes with the risk of HCC and CLD mortality (Supplementary Figure S7b-S7f) nor the interaction effects between specific plasma fatty acids and the genotypes of above SNPs were observed (Figure 4c-4i).

Lines 403-419: Genotyping in the UK Biobank was performed on two closely related purpose-designed arrays: UK BiLEVE and UK Biobank Axiom (53). We selected 5 single-nucleotide polymorphisms (SNPs, Supplementary Table S2): rs738409 [patatin-like phospholipase domain-containing protein 3 (PNPLA3) I148M variant], rs58542926 [transmembrane 6 superfamily member 2 (TM6SF2) E167K], rs641738 [C>T membrane bound O-acyltransferase domain containing 7 (MBOAT7)], rs1260326 [glucokinase regulator (GCKR) P446L] and rs72613567 [17 β -hydroxysteroid dehydrogenase type 13 (HSD17B13):TA], which are associated with cirrhosis and HCC (54). Then, according to the number of risk allele of each SNP, we calculated the polygenic risk scores (PRS) as below:

$$\text{PRS} = 0.266 \times \text{PNPLA3} + 0.274 \times \text{TM6SF2} + 0.065 \times \text{GCKR} + 0.063 \times \text{MBOAT7} \\ - 0.361 \times \text{HSD17B13}$$

We defined genetic risk in thirds: “low risk” (lowest quartile of PRS), “intermediate risk” (second to third quartiles of PRS), and “high risk” (highest quartile of PRS).

We also selected SNPs associated with specific circulating levels of fatty acids at the genome-wide significance (Supplementary Table S4) (20, 21, 55).

Supplementary Table S4. SNPs associated with circulating individual fatty acids (part)

Type of fatty acid	Fatty acid	SNP	Chr	Nearby gene	EA	% Variance explained
n-3 PUFA	ALA	rs174547	11	FADS1	C	1.0
n-3 PUFA	DHA	rs2236212	6	ELOVL2	G	0.7
n-3 PUFA	DPA	rs3734398	6	ELOVL2	C	2.7
n-3 PUFA	DPA	rs174547	11	FADS1	T	8.4
n-6 PUFA	LA	rs174547	11	FADS1	C	7.6-18.1
n-6 PUFA	AA	rs174547	11	FADS1	T	3.7-37.6
MUFA	POA	rs603424	10	SCD/PKD2L1	G	0.3-1.6
MUFA	POA	rs102275	11	FADS1/2	C	0.15-1.0
MUFA	OA	rs102275	11	FADS1/2	C	0.3-2.1
SFA	SA	rs102275	11	FADS1/2	T	0.3-1.2

AA, arachidonic acid; ALA, α -linolenic acid; Chr, chromosome; DPA, docosapentaenoic acid; EA, effect allele; EPA, eicosapentaenoic acid; LA, linoleic acid; OA, oleic acid; PA, palmitic acid; POA, palmitoleic acid; SA, stearic acid.

Supplementary Table S11 (Part):

Fatty acids	Further adjusted for PRS	Further adjusted for FADS1/2 genotype ‡
n-3 PUFA		
Q1	1 (Ref)	1 (Ref)
Q2	0.68 (0.49–0.93)	0.63 (0.46–0.87)
Q3	0.59 (0.42–0.82)	0.55 (0.39–0.77)
Q4	0.48 (0.33–0.69)	0.44 (0.30–0.64)
n-6 PUFA		
Q1	1 (Ref)	1 (Ref)
Q2	0.65 (0.44–0.95)	0.63 (0.43–0.92)
Q3	0.58 (0.36–0.93)	0.57 (0.35–0.90)
Q4	0.53 (0.31–0.90)	0.49 (0.29–0.83)
MUFA		
Q1	1 (Ref)	1 (Ref)
Q2	0.82 (0.51–1.31)	0.83 (0.52–1.32)
Q3	1.14 (0.72–1.79)	1.18 (0.75–1.86)
Q4	1.39 (0.79–2.43)	1.52 (0.87–2.67)
SFA		
Q1	1 (Ref)	1 (Ref)
Q2	1.81 (1.12–2.92)	1.81 (1.12–2.93)

Q3	3.24 (2.07–5.07)	3.25 (2.08–5.08)
Q4	3.50 (2.21–5.52)	3.55 (2.25–5.60)

‡ For n-3 PUFA and n-6 PUFA, rs174547 genotype was further adjusted for; for MUFA and SFA, rs102275 genotype was further adjusted for.

Figure 4. Associations of specific fatty acids with incident HCC and CLD mortality by PRS or specific SNPs

Supplementary Figure S7. Associations of genetic susceptibility and *FADS1/2* genotype with the risk of incident HCC and CLD mortality

Comment 8: It is unclear with the fibrosis risk score is and the rationale for using it. If this is going to be published in a more general journal, a few sentences of background is helpful. This needs to be explained more robustly. Moreover, the main effects of these categories should also be presented for the outcome, as well as the interaction terms.

Reply: Thank you for this suggestion. The reason why we study liver fibrosis is that liver fibrosis is a common pathological feature of CLD, with the formation of a fibrous scar. Fibrosis stage is the most important in terms of determining disease progression to liver-associated complications and mortality (*The Lancet Gastroenterology & Hepatology*, 2016;1:256-260). Early stage of fibrosis is usually asymptomatic, nevertheless, once it progresses to the advanced stage or cirrhosis, the prognosis is always poor. Thus, it is necessary to examine whether the associations between fatty acids and liver related outcomes vary by fibrosis stages. We have added a brief introduction to liver fibrosis on lines 98-103, and presented the main effects and the interaction terms in Supplementary Figure S6 and Figure 3.

Lines 98-103: **Liver fibrosis is a common pathological feature of CLD, with the**

formation of a fibrous scar (22). Fibrosis stage is the most important in terms of determining disease progression to liver-associated complications and mortality (23). Early stage of fibrosis is usually asymptomatic, nevertheless, once it progresses to the advanced stage or cirrhosis, the prognosis is always poor (24). Thus, it is necessary to examine whether the associations between fatty acids and liver related outcomes vary by fibrosis stages.

Lines 183-192: We next investigated a possible interaction between fibrosis stage and specific fatty acids in terms of the outcomes. As expected, higher fibrosis risk was associated with a higher risk of HCC or CLD mortality ($P_{trend} < 0.001$, Supplementary Figure S6). When stratifying associations of specific fatty acids with incident HCC by fibrosis stage (by FIB-4), no interaction was observed between specific fatty acids and fibrosis stage (Figure 3a). As for CLD mortality, significant interactions were found between n-3 PUFA, n-6 PUFA and fibrosis stage ($P_{interaction} = 0.007$ and 0.029 , respectively). Among participants at high fibrosis risk (FIB-4 > 2.67), pronounced associations were found for n-3 PUFA [HR_{1-SD} (95% CI): 0.41 (0.31–0.55)] and n-6 PUFA [HR_{1-SD} (95% CI): 0.34 (0.23–0.51), Figure 3b] compared to those at low fibrosis risk (FIB-4 < 1.30).

Supplementary Figure S6. Associations of FIB-4 categories with the risk of incident HCC and CLD mortality

Figure 3. Associations of specific fatty acids with incident HCC and CLD mortality by fibrosis stage

Comment 9: The sensitivity analyses are more enlightening when all conditions are done together in one model. This should be performed to see how all things together affect the results.

Reply: Thank you for this suggestion. We further conducted sensitivity analyses with further adjustment for the history of lipid-lowering medication, PRS, *FADS1/2* genotype, and mutual adjustment for the remaining plasma fatty acids (n-3 PUFA, n-6 PUFA, MUFA, and SFA) where appropriate. Results were similar to the main analyses in Supplementary Tables S11 and S12. We have added these data on Lines 479-484.

Lines 479-484: **In sensitivity analyses, we further excluded liver-related events in the first two years of follow-up to minimize the possibility of reverse causation. We further adjusted for the history of lipid-lowering medication (Atorvastatin, Crestor, Eptastatin, Fluvastatin, Lescol, Lipitor, Lipostat, Pravastatin, Rosuvastatin, Simvador, Simvastatin, or Zocor), PRS, *FADS1/2* genotype, and remaining plasma fatty acids (n-3 PUFA, n-6 PUFA, MUFA, SFA) where appropriate.**

Supplementary Table S11. Sensitivity analyses of the HRs for the associations of plasma fatty acids levels with incident HCC risk (part)

Supplementary Table S12. Sensitivity analyses of the HRs for the associations of plasma fatty acids levels with CLD mortality (part)

Fatty acids	Exclude the first 2	Further adjusted	for	Further adjusted	Further adjusted	Further adjusted for
-------------	---------------------	------------------	-----	------------------	------------------	----------------------

	years of lipid-lowering medication †	for PRS	for FADS1/2 genotype ‡	remaining plasma fatty acids §
n-3				
PUFA				
Q1				
Q2				
Q3				
Q4				
n-6				
PUFA				
Q1				
Q2				
Q3				
Q4				
MUFA				
Q1				
Q2				
Q3				
Q4				
SFA				
Q1				
Q2				
Q3				
Q4				

† Model was adjusted for age, sex, ethnicity, BMI, waist circumference, Townsend deprivation index, education level, household income, self-reported smoking status, self-reported frequency of alcohol intake, physical activity, diet quality score, baseline hypertension, baseline diabetes, baseline dyslipidemia, total cholesterol level, triglycerides level, total fatty acids level, serum ALT level, serum AST level, and blood platelet level.

‡ For n-3 PUFA and n-6 PUFA, rs174547 genotype was further adjusted for; for MUFA and SFA, rs102275 genotype was further adjusted for.

§ For n-3 PUFA, the remaining plasma fatty acids (n-6 PUFA, MUFA, SFA) were adjusted for; for n-6 PUFA, the remaining plasma fatty acids (n-3 PUFA, MUFA, SFA) were adjusted for; and so on. Total fatty acids level was not adjusted for here.

Comment 10: Supplemental figures S3 and S4 are more important than showing the multiple models. Please only show the fully adjusted model as well as these plots. It would also be more effective to show the main results of the ORs in a figure with plots of the HRs with SEs in the main paper.

Reply: Thank you for this suggestion. Supplementary Figures S3 and S4 (the new numbers are Figures S4 and S5) are Kaplan-Meier curves with fully adjusted, in order to visualize crude survival (incidence) rates at different time points in each quartile of FAs. We also illustrated the fully adjusted results of the HRs with SEs using heatmaps as you suggested (reply to comment 5).

Supplementary Figure S4. Kaplan-Meier survival estimates according to specific fatty acids levels for the probability of incident HCC.

Supplementary Figure S5. Kaplan-Meier survival estimates according to specific fatty acids levels for the probability of CLD mortality.

Comment 11: If you stratify the associations for CLD by HCC, NAFLD, AFLD, and viral hepatitis, are the same associations observed for all pre-disposing conditions of CLD (I recognize that power may be limited if this is done, so it might it be possible)?

Reply: Thank you very much for this comment. In this study, the outcome “CLD mortality” was defined as death from alcoholic liver diseases, liver fibrosis, cirrhosis, and MASLD (ICD-10 codes K70, K74, K75.8, and K76.0). Therefore, we could only stratify them by alcoholic liver diseases (ICD-10 code K70), liver fibrosis or cirrhosis (ICD-10 code K74), and MASLD (ICD-10 codes K75.8, and K76.0). We found that n-3 PUFA and n-6 PUFA were negatively associated with risk of alcoholic liver diseases, liver fibrosis and cirrhosis mortality, while MUFA and SFA were positively associated with risk of alcoholic liver diseases, liver fibrosis and cirrhosis mortality. In addition, we found that SFA was positively associated with MASLD mortality. We have added these data to the revised manuscript on Lines 175-179 and in Supplementary Table S10.

Lines 175-179: **In Supplementary Table S10, n-3 PUFA and n-6 PUFA were negatively associated with risk of alcoholic liver diseases, liver fibrosis and cirrhosis mortality, while MUFA and SFA were positively associated with risk of alcoholic liver diseases, liver fibrosis and cirrhosis mortality. In addition, we found that SFA was positively associated with MASLD mortality.**

Supplementary Table S10. Associations between plasma fatty acids levels and CLD mortality according to the causes.

CLD causes	HRs associated with 1-SD increment in specific plasma fatty acids	P value
ALD (ICD-10 K70)		
n-3 PUFA	0.47 (0.38–0.60)	<0.001
n-6 PUFA	0.23 (0.17–0.32)	<0.001
MUFA	3.08 (2.23–4.24)	<0.001
SFA	2.16 (1.82–2.57)	<0.001
Fibrosis or cirrhosis (ICD-10 K74)		
n-3 PUFA	0.46 (0.32–0.66)	<0.001
n-6 PUFA	0.45 (0.27–0.75)	0.002

MUFA	1.75 (1.04–2.94)	0.036
SFA	2.19 (1.64–2.93)	<0.001
MASLD (ICD-10 K75.8 and K76.0)		
n-3 PUFA	0.69 (0.42–1.13)	0.140
n-6 PUFA	0.53 (0.25–1.13)	0.099
MUFA	1.17 (0.55–2.50)	0.683
SFA	1.85 (1.18–2.89)	0.007

Comment 12: It is notable that non-white participants are removed. What is the justification for this? There are >14,000 non-white individuals removed. Even if the genetic analyses are not possible (which I still think that they are likely possible), it would seem very important to include these in the other analyses, especially since there are a large number of minority groups and there are likely not many other places to study them with regard to these outcomes and FA subclasses. In addition, this adds another point of novelty to this study.

Reply: Thank you for this suggestion. We agree that it is important to include the minority groups. In the revised manuscript, we enrolled non-white participants in the main analyses, and adjusted for ethnicity in the regression models. We only exclude them in genetic analyses, because the PRS is constructed based on GWAS data from European populations. We have added this information to the revised manuscript as follows.

Lines 349-354: The current project included 274,123 participants with valid data on NMR metabolic biomarkers. We then excluded those with end-stage liver diseases (Supplementary Table S1), unavailable genetic or FIB-4 data, and missing covariates, leaving 252,398 participants to analyze the associations. In addition, a subgroup of participants of European descent with available genetic data ($n=236,492$) was extracted only for genetic analyses (Supplementary Figure S1).

Supplementary Figure S1. Flow chart of the study design and analytical strategy.

Replies to Reviewer #2 (Remarks to the Author):

ABSTRACT

Comment 1: Line 52: Redundant of “for MUFAs”. Please remove.

Reply: Thank you very much for this suggestion. We have removed “for MUFAs” in the revised manuscript.

Lines 37: **However, the plasma level of MUFA was only associated with a higher risk of CLD mortality [HR_{Q4vsQ1}: 3.81 (95% CI: 2.03–7.16)] but not HCC.**

INTRODUCTION

Comment 2: Since there are several prospective cohort studies examined the association between fatty acids and HCC, the authors should have laid out a strong rationale for the current study, such as using biomarker based (though they did mention somewhere, yet it did not clear to me).

Reply: Thank you very much for this valuable comment. It is true that there are several prospective cohort studies examined the association between fatty acids and HCC. However, they mainly focus on dietary FAs, and evidence regarding specific dietary FAs in relation to incident HCC remains inconsistent. Notably, great heterogeneity between previous studies may result from the error-prone FA intake assessment using self-report questionnaires or food records. In contrast, objectively assessed circulating FAs are not subject to recall bias and represent the combined effect of dietary intakes and biological processes in vivo, including absorption, synthesis, and metabolism. Thus, as the internal exposures to FAs, circulating FAs levels reflect the metabolic status of FAs. In addition, though there are some studies exploring the relationship between circulating fatty acids and HCC, but they are cross-sectional nature and have relatively small sample sizes (~100). Our study uses objective circulating markers and is a prospective cohort design, with a higher level of evidence. Furthermore, we also consider genetic modification effect using polygenic risk score and specific SNPs.

In addition to HCC, we first explored the association of FAs with CLD deaths (The reason why we chose incident HCC and CLD mortality together as outcomes is explained in the reply to comment 3). Although both outcomes are relatively rare,

sufficient statistical power can be achieved when the sample size is large enough, and UKB has this advantage (~250,000 eligible participants). We have added the above information to the revised manuscript as follows.

Lines 69-91: **Overall, evidence regarding specific dietary fatty acids in relation to incident HCC remains inconsistent.** It has been reported that n-6 PUFA and SFA intake displayed a significantly positive association with HCC risk (8, 9), whereas no risk associations of HCC were detected for PUFA or SFA intake in a European cohort (10). Furthermore, in an analysis of data from 2 large US cohort studies, increasing consumption of MUFA and PUFA, including both n-3 and n-6 PUFA, was associated with a lower risk of developing HCC (11). Correspondingly, in a hospital-based case-control study, inverse associations of MUFA and long-chain n-3 PUFA intake with HCC were found (12). **Nevertheless, information based on dietary recalls inevitably introduces bias, while fatty acids in circulation are an attractive source of biomarkers (13). In addition, the plasma lipidome closely parallels liver lipidome changes (14), and circulating fatty acids have also been found to reflect the composition of liver triglycerides, which makes it easier to explore the role of hepatic fatty acids compared with collecting human tissues (15).** Zhou et al. analyzed serum metabolomics and revealed the deregulation of FA metabolism in HCC and CLD (16). A blood lipidomic study showed the importance of an increased ratio of long-chain n-6 PUFA over n-3 PUFA for HCC risk in both human samples and a mouse tumor model (17). Besides, in 116 subjects of Hispanic descent, plasma concentrations of very long chain SFA and very long chain n-3 PUFA were significantly reduced in patients with HCC (18). **However, these studies have relatively small sample sizes (~100), thus it is necessary to examine the association of HCC with subtypes of circulating fatty acids in large cohorts. Furthermore, although evidence is accumulating on fatty acids and MASLD progression (19), the specific subtype of fatty acids that influences mortality from CLD remains poorly understood.**

Comment 3: Another issue is that they need to justify why they chose 2 endpoints: incident HCC and CLD mortality for the current analysis.

Reply: Thank you for this valuable comment and suggestion. For hepatology research, HCC and CLD mortality are the two outcomes of interest, which represent the advanced state of liver disease. Approximately 70–90% of liver cancer arises in persons with CLD (*Dig Dis.* 2009; 27: 80-92). CLD, including cirrhosis, fibrosis, and alcoholic liver disease, are important precursors of HCC. Additionally, CLD may lead to death itself and is also a significant public health concern. Except hepatitis virus, more HCC and CLD are probably attributable to obesity, T2DM and MASLD, which are closely associated with fatty acids consumption and metabolism. Thus, identifying factors that could reflect the risk of liver cancer and CLD mortality is of interest. Besides, these two outcomes have also been widely focused in the previous studies (*JAMA*, 2023;330:537-546; *Nat Commun*, 2021;12:6388; *Am J Clin Nutr*, 2023;117:278-285; *JHEP Reports*, 2023;5:100819). We have added the above information to the revised manuscript as follows.

Lines 55-61: **Most HCCs occur in the context of chronic liver disease (CLD), such as viral hepatitis, metabolic dysfunction-associated steatotic liver disease (MASLD), and alcoholic fatty liver disease (3). In addition, CLD leads to 2 million deaths worldwide each year and is therefore a notable public health concern (4). Both HCC and CLD deaths represent the advanced state of liver disease. The growing burden related to HCC and CLD emphasizes the importance of identifying people at high risk and taking measures as soon as possible (5, 6).**

Comment 4: Please provide citation(s) for paragraph 2, lines 75-81.

Reply: Thank you for this suggestion. We have added the citation for this paragraph in the revised manuscript as follow.

Reference 7: **Graham C. Burdge, Philip C. Calder, 2014. "Introduction to Fatty Acids and Lipids", Intravenous Lipid Emulsions, <https://doi.org/10.1159/000365423>.**

Comment 5: Line 93: spell out TG the first-time use.

Reply: Thank you for this suggestion. We have spelled out TG as triglycerides in the revised manuscript.

Line 80:and circulating fatty acids have also been found to reflect the composition of liver **triglycerides**.....

Comment 6: Line 102: The NAFLD has new nomenclature, MASLD. Please update this.

Reply: Thank you for this suggestion. NAFLD has been replaced by metabolic dysfunction-associated steatotic liver disease (MASLD) in the revised manuscript.

Line 90: Furthermore, although evidence is accumulating on fatty acids and **MASLD** progression.

DISCUSSION

Comment 7: Line 197: since the analysis in the Finnish cohort study included both cross-sectional and prospective type, your statement in prior lines (195-196) is incorrect. Please modify it.

Reply: Thank you for this comment. We used “few” instead of “no” in the revised manuscript.

Lines 239: To date, **few** prospective population-based studies have examined associations between plasma levels of specific fatty acids and severe liver outcomes.

METHODS

Comment 8: Statistical analysis: (lines 380-382). Please describe more details why you chose quartile scale and how you did it.

Reply: Thank you for this comment. In contemporary epidemiologic practice, continuous variables are typically categorized into quartiles as a means to illustrate the relationship between a continuous exposure and a binary outcome (*The Lancet Planetary Health*, 2019; 3: e478-e490; *Diabetes Care*, 2022; 45(3):564–575). We have added this exploitation to the revised manuscript as follows.

Lines 450-458: **Plasma levels of specific fatty acids were expressed as percentages of plasma total fatty acids, each of which was then divided into quartiles to compare baseline characteristics or risk for incident HCC and CLD mortality across them. To**

describe participant baseline characteristics for each quartile, continuous variables were presented as the mean \pm standard deviation (SD) and categorical variables were presented as percentages. Hazard ratios (HRs) and 95% confidence intervals (CIs) for each quartile of exposure were reported, with the lowest quartile as the reference.

Supplementary Tables S4 and S5. Footnote

Comment 9: Please list all covariates that were adjusted in the “fully adjusted model”.

Reply: Thank you for this suggestion. We have listed all covariates below these tables (Supplementary Tables S11 and S12 are the new table numbers).

Footnote of Supplementary Tables S11 and S12:

† Model was adjusted for age, sex, ethnicity, BMI, waist circumference, Townsend deprivation index, education level, household income, self-reported smoking status, self-reported frequency of alcohol intake, physical activity, diet quality score, baseline hypertension, baseline diabetes, baseline dyslipidemia, total cholesterol level, triglycerides level, total fatty acids level, serum ALT level, serum AST level, and blood platelet level.

‡ For n-3 PUFA and n-6 PUFA, rs174577 genotype was further adjusted for; for MUFA and SFA, rs102275 genotype was further adjusted for.

§ For n-3 PUFA, the remaining plasma fatty acids (n-6 PUFA, MUFA, SFA) were adjusted for; for n-6 PUFA, the remaining plasma fatty acids (n-3 PUFA, MUFA, SFA) were adjusted for; and so on. Total fatty acids level was not adjusted for here.

REVIEWERS' COMMENTS

Reviewer #1 (Remarks to the Author):

The authors did an excellent job responding to the questions and concerns provided in the reviews. Thank you for the careful response to the comments that were provided.

Reviewer #2 (Remarks to the Author):

See my comments to the Editors

RESPONSE TO REVIEWERS' COMMENTS

Reply to Reviewer #1:

The authors did an excellent job responding to the questions and concerns provided in the reviews. Thank you for the careful response to the comments that were provided.

Reply: We thank the reviewer for the recommendations to improve the clarity in the methods and results, and review of our manuscript.

Replies to Reviewer #2 (Remarks to the Author):

See my comments to the Editors

Reply: We thank the reviewer's suggestions in strengthening the manuscript and narrative.